# Increasing childhood illnesses (diarrhea and fever) and decreasing care-seeking practices in Nepal: Insights from three most recent Demographic and Health Surveys (2011, 2016 and 2022)

**Resham B. Khatri**[1,2]\*, **Rolina Dhital**[3], **Sabita Tuladhar**[4], **Ravi Kanta Mishra**[5,6], **Yibeltal Assefa**[2]

1 Health Social Science and Development Research Institute, Kathmandu, Nepal, 2 School of Public Health, University of Queensland, Brisbane, Australia, 3 Health Action and Research, Kathmandu, Nepal, 4 Nepal Public Health Association, Kathmandu, Nepal, 5 Ministry of Health and Population, Kathmandu, Nepal, 6 Department of Global Public Health and Primary Care, BCEPS, University of Bergen, Bergen, Norway

\* rkchettri@gmail.com

## Abstract

Nepal has notably reduced major childhood illnesses like diarrhea and acute respiratory infections via community-based child health programs. This study analyzes trends and factors influencing the prevalence of fever and diarrhea in children under five, along with care-seeking behaviors, using data from the Nepal Demographic and Health Surveys of 2011, 2016, and 2022. Between 2011 and 2022, fever prevalence increased from 19% to 23%, while care-seeking for diarrhea and fever declined slightly (diarrhea: 62% to 57%; fever: 80% to 78%). Care-seeking at private health facilities increased for both illnesses (diarrhea: 37% to 42%; fever: 54% to 67%). Key determinants varied by condition and region. Diarrhea prevalence was lower in children aged 36–59 months but higher in Bagmati and Karnali provinces and in Hill and Terai regions. Care-seeking for diarrhea was less likely for first-born children and more likely in Lumbini province and among children of native Maithili-speaking mothers. Fever prevalence was higher among children aged six months to four years, those born to native Nepali-speaking mothers, and second-born children. Care-seeking for fever was lower in children of mothers facing multiple disadvantages but higher in Madhesh province. Care at private health facilities for fever was more common among children with fewer maternal disadvantages, living in urban areas, from Lumbini or Madhesh provinces, or with native Maithili-speaking mothers. The increasing prevalence of childhood illnesses combined with decreasing care-seeking practices underscores a major public health challenge. Most caregivers/mothers opting for private health facilities suggests trust or access issues with public services and government facilities. Programs should focus on recruiting health care providers

**Data availability statement:** All relevant data are available in the paper and its Supporting Information files.

**Funding:** The authors received no specific funding for this work.

**Competing interests:** The authors have declared that no competing interests exist.

with good understanding of local languages and cultures, focus on province-specific health challenges, and enhance health communication in local languages, especially in the Terai region, to reduce the burden of childhood illnesses and improve care-seeking practices in Nepal.

## Introduction

Over the past three decades, Nepal has achieved substantial progress in reducing child mortality. The under-five mortality rate (U5MR) decreased from 152 to 32 per 1,000 live births between 1996 and 2022 [1–5]. While the U5MR is declining, significant disparities continue to exist between the wealthiest and the poorest groups. In 2006, the ratio of U5MR between the poorest wealth quintile (U5MR:98) and the wealthiest quintile (U5MR:47) was 2.08 [5]. By 2022, this gap had widened, with the U5MR in the poorest quintile rising to 53, while in the wealthiest quintile, it dropped to 16, resulting in a ratio of 3.31 [5]. Households with intersectional disadvantages and comparative privileges might face even wider equity gaps, but there is little evidence in literature.

Nepal has taken steps to narrow the equity gap and remains dedicated to achieving universal access to quality essential health services, including those for children. The National Strategy for Reaching the Unreached (2016–2030) aims to achieve universal health coverage (UHC) for marginalized populations by tackling both demand- and supply-side challenges within the health system [6]. Moreover, the female community health volunteers (FCHVs) have also been instrumental in narrowing equity gaps by reaching the underserved population in communities [7]. With around 52,000 FCHVs nationwide, they actively contribute to national immunization programs and child health campaigns related to child survival which contribute to the reduction of U5MR [8]. Since its inception in the 1990s, the Community Based-Integrated Management of Neonatal and Childhood Illnesses (CB-IMNCI) Program has been one of the most successful programs for treating major fatal childhood diseases (diarrhea, acute respiratory infection (ARIs), malaria, measles, and malnutrition) [9,10].

Despite the policies and programmatic provisions, the equity gap continues to exist with a high burden of childhood illnesses. Care-seeking practices for childhood illnesses also remain poor. While treatment for childhood illnesses such as fever, ARIs, and diarrhea is provided free of charge in public health facilities (HFs) as part of Basic Health Services (BHS), many people still choose to seek care at private HFs, where treatments are not free [11]. This is because free treatment is not universally available in public HFs in Nepal. For instance, in mixed health systems, private health facilities provide management of childhood diseases on a fee-for-service basis, even though these services are freely available through public funding.

The Nepal Health Facility Survey (NHFS) 2021 revealed that 20% of sick children attending public HFs still paid for treatment [12]. Moreover, services included in the BHS package are not covered by the national health insurance program [13]. The tendency of people from disadvantaged seeking care from private HFs further exacerbates inequities due to higher costs of care and a lack of financial protection [14].

Health equity is influenced by multiple and interconnected factors such as structural, intermediary and health system factors. Broadly, structural and societal issues such as education, ethnicity, wealth, and systemic marginalization can create structural inequities that require structural reforms and changes because they lie beyond the control of health systems [15–17]. On a more immediate level, intermediary factors influence daily activities such as living and working conditions and infrastructure and are modifiable factors through multisectoral actions. These factors can be modified through development policies to enhance access to resources including health services. Meanwhile, the health system itself plays a role by improving the readiness and quality of healthcare services to promote equitable health outcomes. Individuals and families face influences from both broad societal structures and immediate conditions that impact their health. Structural factors, which are embedded in society and require political action to change, are intertwined with intermediary factors that can be improved through coordinated efforts across different sectors [18]. These overlapping factors interact with various aspects of identity, resulting in complex layers of marginalization that hinder access to child health services including health care seeking common childhood illnesses for fever and diarrhea.

A large body of literature has documented childhood illnesses and care practices, primarily further analysis of Nepal Demographic and Health Surveys (NDHSs) focusses on determinants of child health illnesses in Nepal, focussed on associated determinants of prevalence and care seeking practices [19–22]. A qualitative study explored the relationship between childhood diarrhea and factors such as the child's age, gender, hand-washing behavior, nutritional status, mother's education, water and sanitation conditions, healthcare services, cultural and societal values, and household income [23]. Community-based studies reported that respondents had a fair perception of childhood fever, and about half had poor practices regarding its management [24,25]. Evidence showed that on trends in care-seeking practices for fever, cough, and diarrhea among children using nationally representative household survey data from 2001 to 2016, reported a declining prevalence of these symptoms alongside low care-seeking practices [26]. However, there have been limited studies on the recent trends and determinants of care seeking practices for common childhood illnesses (e.g., fever and diarrhea), especially the prevalence and place of care seeking those symptoms among children from the households with multiple forms of marginalization status (e.g., children living in the families with low wealth status and illiterate mothers and under privileged ethnic groups). These groups have been experiencing higher child mortality rates and low access to child health services when their children face common childhood illnesses such as diarrhoea, fever and pneumonia.

In the context of Nepal, it is essential that all children have access to quality healthcare, with a particular focus on populations experiencing high under-five mortality rates and addressing health disparities more effectively. It is vital to assess the scale of these inequities in order to design targeted strategies and programs that provide fair access to child health services for those who need them the most. Tackling these disparities is critical for updating and refining policies, strategies, and programs within Nepal's decentralized health system. A thorough understanding of inequity trends and their causes will help policymakers design interventions for the most vulnerable groups, such as women facing multiple forms of marginalization. Consequently, this study aims to examine the equity gaps in child health services in Nepal, in particular investigate the trends in the prevalence of, care-seeking for, and place of care-seeking for diarrhea and fever across the three most recent NDHSs (2011, 2016, and 2022) [3–5], and examine the determinants of the prevalence of fever and diarrhea, as well as care seeking practices and place of care-seeking in the NDHS 2022 [5]. The findings of this study could inform policy and programs to address these childhood illnesses and care-seeking practices among children under five years.

## Methods

### Data sources

This study used data from the three most recent NDHSs (2011, 2016, and 2022) [3–5]. Trend analyses were conducted using data from the NDHS 2011 (N = 5,140) [3], NDHS 2016 (N = 4,887) [4], and NDHS 2022 (N = 5,040) [5] among children under five years old who had experienced childhood illnesses (diarrhea or fever) in the two weeks prior to the

surveys. Determinants were identified using data from the NDHS 2022 for children with diarrhea (n = 524) and fever (n = 1,159), as well as those who received treatment for diarrhea (n = 300) and fever (n = 905).

## Study variables

**Outcome variables.** This study focused on examining the prevalence of childhood diarrhea and fever, care-seeking practices, and place of care-seeking for these illnesses based on the responses reported by the mothers. The study included the following key variables: whether the child had diarrhea or a fever in the two weeks before the survey (Yes/No), whether treatment was sought for diarrhea or fever (Yes/No), and place of care seeking (public/private).

**Independent variables.** Independent variables considered in the study included various background characteristics of children (child's sex, age, and birth order) and mothers (maternal age, education level, religion, ethnicity, wealth quintile, marginalization status, province, place of residence, ecological region, occupation, and native language) [27]. The Government of Nepal has categorized different ethnicities into six broader categories: i) Dalits, ii) disadvantaged indigenous, iii) disadvantaged non-Dalit Terai caste groups, iv) religious minorities (Muslims), v) relatively advantaged indigenous groups, and vi) upper caste groups (advantaged groups include Brahmin and Chhetri) [28–30]. Referring to previous studies, a variable of marginalization status was created using three background variables of mothers: education, wealth status, and ethnicity [14,30,31]. In the first step, the six broader ethnic groups were categorized into two broader groups according to their comparative privileges: disadvantaged ethnicities (including Dalit, Muslims, Terai caste, and disadvantaged Janajatis) and advantaged ethnicities (including Brahmin/Chhetri and advantaged Janajatis) [28,30]. Education was classified as either illiterate (those unable to read and write) or literate (those who could read and write or had at least primary education). Wealth status of households was split into two categories: lower wealth status (the lowest two quintiles) and upper wealth status (top three quintiles). Based on literature and first author's previous work, new variable "marginalization status" was created using three background variables: education, wealth status, and ethnicity [14,30–35]. By combining three variables with two categories, we created a new variable (marginalization status) with eight categories. We merged three categories with at least one form of marginalization into one category, and the same was done for two forms of marginalization in the next category. So, the new variable with multiple forms of marginalization status had four categories: triple, double, single, and no marginalization [30,34].

## Statistical analyses

The analysis used two distinct methods to examine trends and the factors influencing the outcome variables. First, we performed a descriptive trend analysis (using data from the three most recent NDHSs 2011, 2016, and 2022), presenting proportions in the figures and assessing the significance of these trends using a proportion test (Stata command "prtesti"). To identify factors associated with prevalence of care seeking of child health service utilization, we conducted univariable analyses (sample distribution), bivariable analyses (examining the relationship between each independent variable and each outcome), and multivariable logistic regression analyses (adjusted models). We conducted survey logistic regression models to identify determinants for each outcome variable considers the sampling design of the survey, including stratification, clustering, and weighting [19,20,36,37]. The DHS dataset includes survey weights which are applied to correct for unequal probabilities of selection and non-response biases. Clustering, which arises when data are collected from groups or clusters, is addressed by adjusting standard errors to reflect the intra-cluster correlation. Stratification (such as province) involves dividing the population into subgroups used in models. Furthermore, prior to running the final regression models, multicollinearity was checked, and independent variable (ecoregion) with a variance inflation factor (VIF) ≥ 5 were excluded. wealth status, ethnicity, and education were also excluded in the adjusted regression model as these variables were highly correlated with variable multiple marginalization. Adjusted odds ratios (AOR) with 95% confidence intervals (CI) were reported for all independent variables with p-values < 0.05 [36]. The study reported the AOR with a 95% confidence interval (CI), and the statistical significance level was set at p < 0.05 (two-tailed). All analysis output included

in this study are weighed estimates (otherwise indicated). All analyses were conducted using the "svy" command function which adjust clustering effect of survey design by adjusted cluster variable along with stratification and sampling weights and ensure correct variance estimation and inference. The goodness of fit was assessed using the Hosmer-Lemeshow test, with non-significant results (p > 0.05) indicating adequate fit. Statistical significance was set at p < 0.05 (two-tailed) to identify determinants associated with the outcome variables. All findings are estimates unless otherwise stated. Analyses were conducted using the "svy" command function in Stata 20 (StataCorp, 2023), which accounted for clustering effects.

### Research ethics

We used publicly accessible, de-identified datasets from the Demographic and Health Survey program (https://dhspro-gram.com/data/available-datasets.cfm). For each round of the NDHS, the survey received ethical approval from the ICF Institutional Review Board in the USA and the Nepal Health Research Council. Given that this research involved only a secondary analysis of completely anonymized data, there was no need for additional ethical clearance. The authors received approval to access and utilize the de-identified data.

## Results

### Trends childhood illnesses from 2011 to 2022

Following two figures (Fig 1 and 2) provide the prevalence, care-seeking practices, and place of care-seeking for diarrhea and fever among children under five years old in Nepal. Particularly, Fig 2 illustrates the trends in the prevalence, care-seeking (treatment) for diarrhea, and the place of care seeking for the years 2011, 2016, and 2022. The vertical line pattern indicates a statistically significant change (p < 0.05) between the first two NDHSs (2011 and 2016), the horizontal line pattern indicates a statistically significant change between the second two NDHSs (2016 and 2022), and the thick black outline denotes a significant change between 2011 and 2022. Nationally, the overall prevalence of diarrhea declined from 14% in 2011 to 10% in 2022. However, after a decrease from 14% in 2011 to 8% in 2016, the prevalence of diarrhea rose again to 10% in 2022. Diarrhea prevalence decreased across all marginalization groups from 2011 to 2016 but significantly increased from 2016 to 2022 among those with double disadvantages and no disadvantages. From 2011 to 2022, diarrhea prevalence significantly declined among all marginalization groups, except for those with no disadvantages. The lowest prevalence of diarrhea was observed among children with no disadvantages.

Nationally, care-seeking practices for diarrhea did not show a significant change from 2011 to 2022, but it declined significantly in the most recent period (2016–2022), from 65% to 57%. Care-seeking for diarrhea decreased from 2011 to 2022 among children with a single disadvantage, but no significant changes were observed among other marginalized groups (Fig 2).

The percentage of children with diarrhea who sought treatment at public HFs declined consistently from 24.9% in 2011 to 15.6% in 2022. The percentage of children with diarrhea seeking care in the private HFs increased from 37.1% in 2011 to 47.7% in 2016 but then decreased slightly to 41.6% in 2022. The proportion of children with diarrhea who did not seek treatment decreased from 38% in 2011 to 35% in 2016, before rising to 42.7% in 2022 (Fig 2).

Fig 1 illustrates the trends in the prevalence, care-seeking practices, and place of care-seeking for fever among children under five years old, categorized by marginalization status. Nationally, the prevalence of fever increased from 19% in 2011 to 23% in 2022. The lowest prevalence of fever was observed among children with triple disadvantages, while the highest prevalence was seen among those with no disadvantages. The prevalence of fever increased from 2011 to 2022 across all marginalization categories, except for those with a single disadvantage. The gap in fever prevalence based on marginalization status remained relatively unchanged from 2011 to 2022 (Fig 1).

Nationally, the percentage of children with fever seeking treatment increased from 72% in 2011 to 78% in 2022. Treatment for fever also rose significantly among children with triple disadvantages, from 59% in 2011 to 79% in 2022. No other marginalization groups showed significant changes from 2011 to 2022, although care-seeking for fever declined

**A:** Prevalence of childhood fever

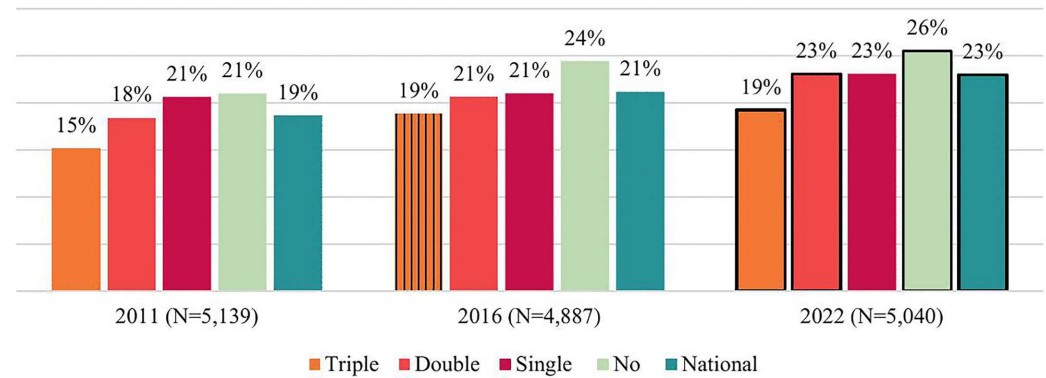

**Note:** Vertical lines indicate significant change from 2011–2016, and thick black outlines from 2011–2022 at significance level p<0.05.

**B:** Care seeking for childhood fever

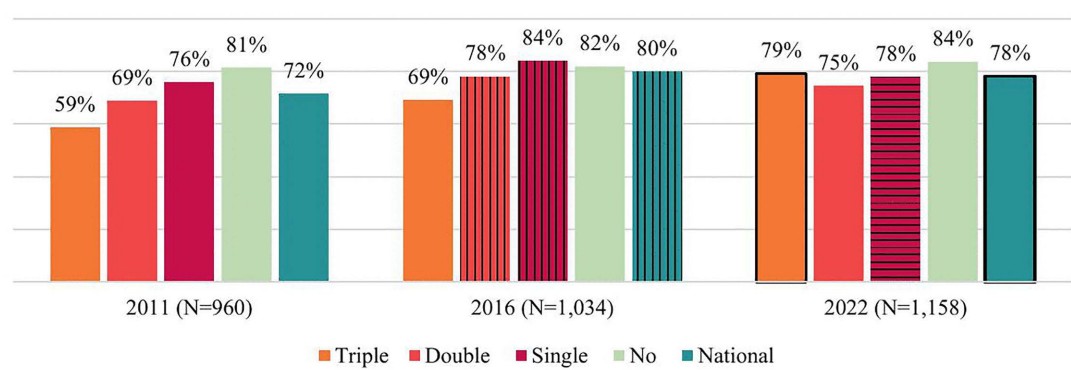

**Note:** Bars with vertical lines show significant change from 2011–2016, horizontal lines from 2016–2022, and thick black outlines from 2011–2022 at significance level p<0.05

**C:** Place of care seeking for childhood fever

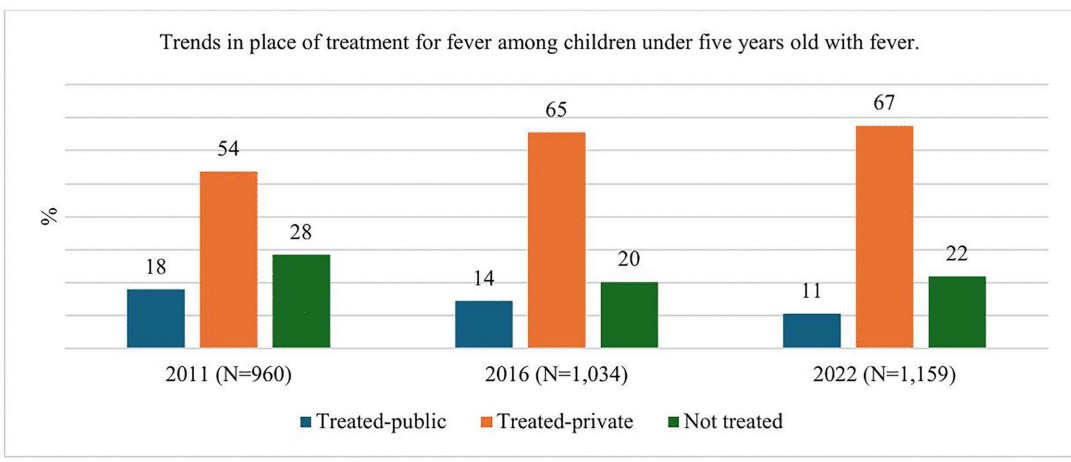

**Fig 1. Trends of fever prevalence, care-seeking practices, and place of care-seeking among children under five in Nepal.**

**A:** Prevalence of childhood diarrhea

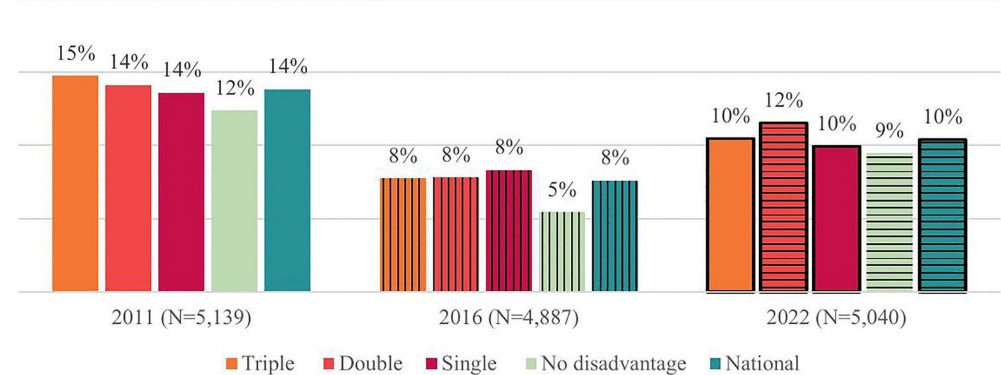

**Note:** Vertical lines indicate significant change from 2011–2016, horizontal lines from 2016–2022, and thick black outlines from 2011–2022 at significance level p<0.05.

**B:** Care seeking for childhood diarrhea

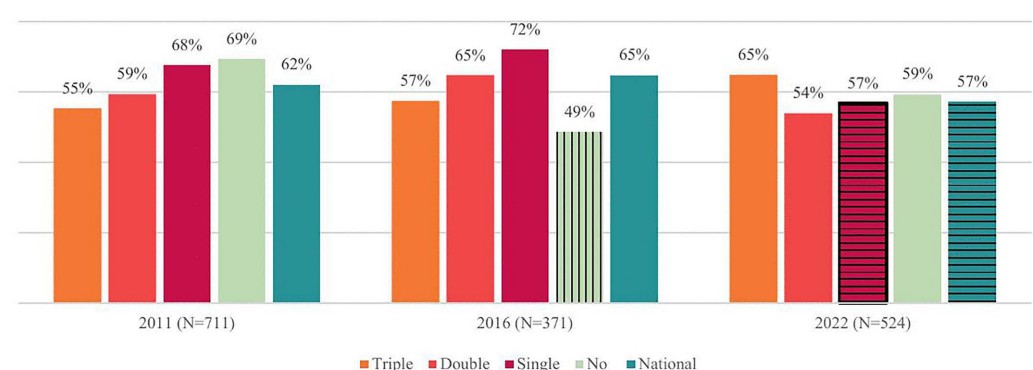

**Note:** Bars with vertical lines show significant change from 2011–2016, horizontal lines from 2016–2022, and thick black outlines from 2011–2022 at significance level p<0.05.

**C:** Place of care seeking for childhood diarrhea

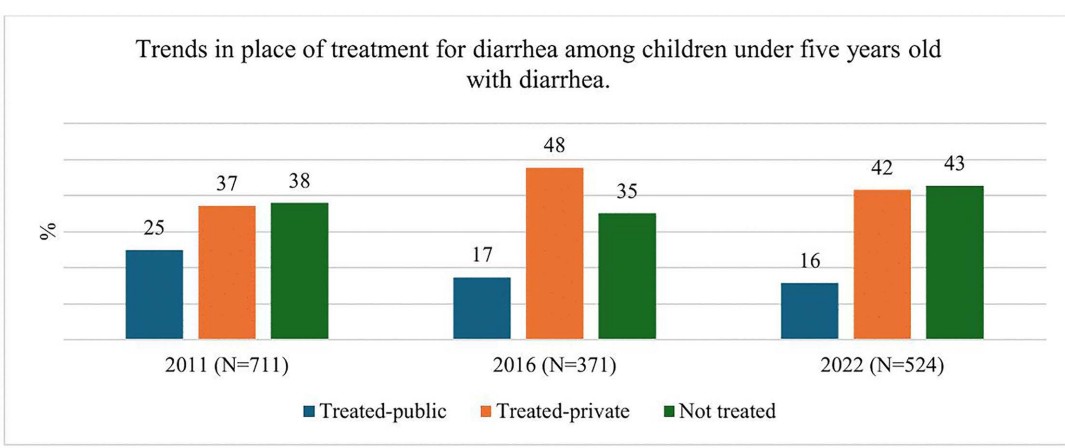

**Fig 2. Trends in prevalence of diarrhea, care-seeking practices, and place of care seeking among children under five years.**

significantly in the most recent period (2016–2022) among children with a single disadvantage, dropping from 84% to 78% (Fig 1).

The proportion of children seeking treatment in private HFs increased steadily from 54% in 2011 to 67% in 2022. Meanwhile, the proportion of children treated in public HFs declined from 18% in 2011 to 11% in 2022. The percentage of children not seeking treatment decreased from 28% in 2011 to 20% in 2016, before rebounding slightly to 22% in 2022 (Fig 1).

## Descriptive analysis of sample included in the analysis

Table 1 presents the distribution of children under 5 from the NDHS 2022, stratified by background variables. Of the 5,040 children included in the analysis, the largest proportions were male (52.4%), aged 36–47 months (20.8%), born to mothers aged 20–29 years (65.2%), Hindu (83.7%), of Janajati (indigenous) ethnicity (28.1%), born to mothers with secondary education (38.8%), from the lowest wealth quintile (24.1%), with a single disadvantage (42%), from Madhesh province (26.8%), living in urban areas (65%), born to native Nepali speakers (47.3%), and firstborn (40.7%).

Overall, 10.4% of the children had diarrhea in the two weeks preceding the survey. Children had a higher prevalence of diarrhea if they were aged 6–11 months (17.5%) and if they were in the middle wealth quintile (12.7%) compared to the national average and to their respective reference groups [see S1 Table]. Additionally, 23% of children had a fever in the two weeks before the survey. The prevalence of fever was significantly higher among children aged 6–12 months (27.3%), those from Karnali province (28%), those from the Hill ecoregion (26.1%), and those born to native Nepali- speaking mothers (26.1%)[see S1 Table].

Nearly three in five (57.2%) children who had diarrhea sought care. The proportion of those who sought care was significantly lower among children with a second birth order (47.6%) compared to the national average or the reference groups [see S2 Table]. Additionally, 78.1% of children with fevers sought care. The proportion of children who sought care was significantly lower among those of Newar ethnicity (52.3%), those in the lowest wealth quintile (68.6%), those from Sudurpaschim province (70.5%), those from the Hill ecoregion (69.4%), and those born to mothers who were native Nepali speakers (75.2%) [see S2 Table].

Overall, for children who had diarrhea and sought care, 72.7% attended private HFs. Children had significantly higher levels of care seeking in private HFs if they were born to mothers of Madhesi ethnicity (87.9%), were in the fourth wealth quintile (87.2%), were from Lumbini province (86.7%), were from the Terai ecoregion (85.1%), and were born to mothers who were Maithili speakers (87.1% [see S3 Table]. Additionally, about 8 out of 10 children with fevers who sought care attended private HFs (80.4%). The proportion of children who sought care at private HFs was significantly higher among those born to Muslim women (98%), those in the highest wealth quintile (98%), those with no disadvantages (90%), those living in Madhesh province (92%), those in urban areas (85.4%), those in the Terai ecoregion (92.3%), those born to native Maithili speakers (93.9%), and those who were second in the birth order (84.9%) [see S3 Table].

## Determinants of prevalence of diarrhea and fever

Table 2 presents the determinants prevalence of diarrhea in the two weeks preceding the survey among children under five years old [Table 2] [see bivariable analysis, S4 Table]. Compared to children under six months, the odds of diarrhea were lower in older children, specifically those aged 36–47 months (AOR = 0.51; 95% CI:0.33–0.78) and 48–59 months (AOR = 0.62; 95% CI:0.42–0.93). Children from Bagmati (AOR = 2.15; 95% CI:1.35–3.42) or Karnali provinces (AOR = 1.68; 95% CI:1.07–2.65) and those from the Hill (AOR = 1.54; 95% CI:1.09–2.17) or Terai regions (AOR = 2.68; 95% CI: 1.73–4.14) had higher odds of diarrhea compared to children from Sudurpashchim province and the Mountain region, respectively. Similarly, the odds were higher among children aged 6–47 months compared to those under six months: 6–11 months (AOR = 1.69; 95% CI:1.21–2.37), 12–23 months (AOR = 1.50; 95% CI:1.14–1.99), 24–35 months (AOR = 1.51; 95% CI: 1.11–2.05), and 36–47 months (AOR = 1.49; 95% CI:1.11–1.99). Additionally, the odds of fever were higher among children born to native Nepali-speaking mothers (AOR = 1.42; 95% CI:1.13–1.78) and among second-born

**Table 1. Distribution of children under five years old, by background variables, NDHS 2022.**

| Maternal/child characteristics | N = 5,040 | Percent | 95%CI |
|---|---|---|---|
| **Sex of child** | | | |
| Male | 2,639 | 52.4 | 50.6–54.2 |
| Female | 2,401 | 47.6 | 45.8–49.4 |
| **Child age in months** | | | |
| <6 | 533 | 10.6 | 9.7–11.5 |
| 6–11 | 434 | 8.6 | 7.7–9.6 |
| 12–23 | 959 | 19.0 | 17.9–20.2 |
| 24–35 | 1,066 | 21.1 | 19.9–22.4 |
| 36–47 | 1,048 | 20.8 | 19.6–22.0 |
| 48–59 | 1,000 | 19.8 | 18.7–21.0 |
| **Maternal age in years** | | | |
| <20 | 994 | 19.7 | 18.3–21.2 |
| 20–29 | 3,286 | 65.2 | 63.6–66.8 |
| Thirty and above | 761 | 15.1 | 13.8–16.4 |
| **Religion** | | | |
| Hindu | 4,218 | 83.7 | 80.8–86.2 |
| Other (e.g., Buddhism, Islam) | 822 | 16.3 | 13.8–19.2 |
| **Ethnicity** | | | |
| Brahmin | 374 | 7.4 | 6.1–8.9 |
| Chhetri | 873 | 17.3 | 15.3–19.5 |
| Madheshi | 1,005 | 19.9 | 17.1–23.1 |
| Dalit | 935 | 18.6 | 16.1–21.3 |
| Janajati | 1,417 | 28.1 | 25.3–31.1 |
| Newar | 119 | 2.4 | 1.6–3.4 |
| Muslim | 317 | 6.3 | 4.2–9.2 |
| **Maternal education** | | | |
| No education | 1,133 | 22.5 | 20.1–25 |
| Basic | 1,737 | 34.5 | 32.5–36.5 |
| Secondary | 1,955 | 38.8 | 36.4–41.3 |
| Higher | 215 | 4.3 | 3.5–5.2 |
| **Wealth quintile** | | | |
| Poorest | 1,213 | 24.1 | 21.5–26.8 |
| Poorer | 1,037 | 20.6 | 18.4–22.9 |
| Middle | 1,048 | 20.8 | 18.9–22.9 |
| Richer | 969 | 19.2 | 17.1–21.6 |
| Richest | 774 | 15.4 | 13.3–17.7 |
| **Marginalization status** | | | |
| Triple | 615 | 12.2 | 10.2–14.5 |
| Double | 1,510 | 30.0 | 27.6–32.4 |
| Single | 2,118 | 42.0 | 39.5–44.5 |
| No | 797 | 15.8 | 13.8–18.0 |
| **Province** | | | |
| Koshi | 859 | 17.0 | 15.3–18.9 |
| Madhesh | 1,352 | 26.8 | 24.4–29.4 |
| Bagmati | 814 | 16.2 | 14.2–18.3 |
| Gandaki | 331 | 6.6 | 5.6–7.7 |

*(Continued)*

**Table 1.** (Continued)

| Maternal/child characteristics | N = 5,040 | Percent | 95%CI |
|---|---|---|---|
| Lumbini | 862 | 17.1 | 15.4–19.0 |
| Karnali | 371 | 7.4 | 6.6–8.3 |
| Sudurpashchim | 451 | 8.9 | 8.0–10.0 |
| **Residence** | | | |
| Urban | 3,276 | 65.0 | 62.7–67.3 |
| Rural | 1,764 | 35.0 | 32.7–37.3 |
| **Ecological region** | | | |
| Mountain | 317 | 6.3 | 4.6–8.6 |
| Hill | 1,744 | 34.6 | 30.8–38.6 |
| Terai | 2,979 | 59.1 | 55.2–62.9 |
| **Native language** | | | |
| Nepali | 2,383 | 47.3 | 43.8–50.8 |
| Maithili | 1,006 | 20.0 | 16.1–24.5 |
| Bhojpuri | 466 | 9.2 | 6.2–13.5 |
| Other (Magar, Tharu) | 1,185 | 23.5 | 20.6–26.7 |
| **Birth order** | | | |
| First | 2,050 | 40.7 | 39.0–42.4 |
| Second | 1,715 | 34.0 | 32.5–35.5 |
| Third or higher | 1,275 | 25.3 | 23.5–27.1 |

children (AOR = 1.22; 95% CI:1.02–1.45), compared to children born to mothers who spoke other languages and those who were first-born, respectively.

## Determinants of care-seeking for childhood diarrhea and fever

Table 3 presents the determinants of care-seeking practices for diarrhea among children under five years old who had diarrhea in the two weeks prior to the survey [see S5 Table]. The odds of care-seeking for diarrhea were lower among first-born children (AOR = 0.49; 95% CI:0.30–0.78) compared to those who were second-born. Children with one or more disadvantages had lower odds of seeking care for fever compared to those with no disadvantages. For example, children with triple disadvantages were 0.42 times as likely to seek treatment for fever (AOR = 0.42; 95% CI: 0.19–0.90) compared to children with no disadvantages. The study also observed higher odds of care-seeking for children living in Madhesh province (AOR = 2.91; 95% CI:1.47–5.75) compared to those in Koshi province.

## Determinants of care-seeking for diarrhea in private HFs

Table 4 presents the determinants of care-seeking for diarrhea among children under five years old who sought care at private HFs in the two weeks prior to the survey [see S6 Table]. The odds of seeking care in private HFs were higher for children from Lumbini province (AOR = 3.48; 95% CI:1.05–11.55), and for children born to mothers who were native Maithili speakers (AOR = 4.38; 95% CI:1.15–16.76) or native speakers of other languages (AOR = 3.06; 95% CI:1.24–7.56), compared to children from Sudurpashchim province and children born to native Nepali-speaking mothers, respectively. Similarly, higher odds of care-seeking for fever in private HFs were found among children born to mothers with fewer disadvantages, including those with a single disadvantage (AOR = 4.58; 95% CI:2.03–10.30) or no disadvantage (AOR = 10.98; 95% CI: 4.22–28.57), compared to children with triple disadvantages. Place of residence was also associated with care-seeking for fever in private HFs. Specifically, the study found higher odds of care-seeking among

**Table 2. Prevalence of diarrhea and fever among children aged under five years, NDHS 2022.**

| Characteristics | Categories | Diarrhea | | Fever | |
|---|---|---|---|---|---|
| | | AOR | 95%CI | AOR | 95%CI |
| **Child age in months** | <6 | 1.00 | | 1.00 | |
| | 6–11 | 1.53 | 1.00–2.35 | 1.69** | 1.21–2.37 |
| | 12–23 | 1.09 | 0.76–1.58 | 1.5** | 1.14–1.99 |
| | 24–35 | 0.77 | 0.51–1.15 | 1.51** | 1.11–2.05 |
| | 36–47 | 0.51** | 0.33–0.78 | 1.49** | 1.11–1.99 |
| | 48–59 | 0.62* | 0.42–0.93 | 1.10 | 0.81–1.49 |
| **Sex of the child** | Male | 1.00 | | 1.00 | |
| | Female | 0.95 | 0.76–1.17 | 0.93 | 0.8–1.08 |
| **Maternal age in years** | <20 | 1.00 | | 1.00 | |
| | 20–29 | 0.77 | 0.56–1.05 | 0.97 | 0.79–1.20 |
| | Thirty and above | 0.72 | 0.45–1.16 | 0.91 | 0.66–1.24 |
| **Religion** | Hindu | 1.00 | | 1.00 | |
| | Other (e.g., Buddhism, Islam) | 0.86 | 0.62–1.18 | 1.00 | 0.79–1.27 |
| **Marginalization status** | Triple | 1.00 | | 1.00 | |
| | Double | 1.06 | 0.72–1.57 | 1.19 | 0.87–1.63 |
| | Single | 0.80 | 0.54–1.19 | 1.16 | 0.86–1.58 |
| | No | 0.66 | 0.39–1.14 | 1.19 | 0.82–1.72 |
| **Province** | Koshi | 1.17 | 0.75–1.83 | 1.08 | 0.78–1.48 |
| | Madhesh | 0.78 | 0.44–1.39 | 0.72 | 0.49–1.07 |
| | Bagmati | 2.15** | 1.35–3.42 | 0.82 | 0.6–1.10 |
| | Gandaki | 1.06 | 0.62–1.8 | 1.02 | 0.72–1.43 |
| | Lumbini | 0.95 | 0.61–1.46 | 0.95 | 0.71–1.27 |
| | Karnali | 1.68* | 1.07–2.65 | 1.08 | 0.82–1.42 |
| | Sudurpashchim | 1.00 | | 1.00 | |
| **Residence** | Urban | 1.00 | | 1.00 | |
| | Rural | 0.85 | 0.65–1.12 | 0.92 | 0.77–1.1 |
| **Ecological region** | Mountain | 1.00 | | | |
| | Hill | 1.54* | 1.09–2.17 | | |
| | Terai | 2.68*** | 1.73–4.14 | | |
| **Native language** | Nepali | 0.97 | 0.71–1.33 | 1.42** | 1.13–1.78 |
| | Maithili | 0.99 | 0.64–1.52 | 1.41 | 0.97–2.04 |
| | Bhojpuri | 0.93 | 0.45–1.94 | 1.12 | 0.73–1.73 |
| | Other | 1.00 | | 1.00 | |
| **Birth order** | First | 1.00 | | 1.00 | |
| | Second | 1.09 | 0.87–1.36 | 1.22* | 1.02–1.45 |
| | Third or higher | 1.02 | 0.71–1.45 | 1.11 | 0.88–1.39 |

*Note: Significance at * p<0.05, ** p<0.01, *** p<0.001.*

children from Madhesh (AOR = 3.64; 95% CI:1.52–8.72) and Lumbini (AOR = 4.69; 95% CI:2.06–10.67) provinces, compared to those from Koshi province, and among children from urban areas (AOR = 2.28; 95% CI:1.48–3.50), compared to those from rural areas. Additionally, compared to children of mothers who were native Nepali speakers, children born to Maithili-speaking mothers (AOR = 5.65; 95% CI:2.39–13.34) and those born to speakers of other languages (AOR = 1.82; 95% CI:1.00–3.29) also had higher odds of seeking treatment in private HFs.

**Table 3. Determinants of care-seeking among children who had diarrhea two weeks prior to the survey, NDHS 2022.**

| Characteristics | Categories | Diarrhea | | fever | |
|---|---|---|---|---|---|
| | | AOR | 95%CI | AOR | 95% CI |
| **Child age in months** | <6 | 1.00 | | 1.00 | |
| | 6–11 | 0.96 | 0.45–2.08 | 1.19 | 0.56–2.54 |
| | 12–23 | 0.96 | 0.49–1.87 | 1.01 | 0.50–2.04 |
| | 24–35 | 0.62 | 0.28–1.36 | 1.03 | 0.50–2.09 |
| | 36–47 | 0.70 | 0.30–1.62 | 0.90 | 0.45–1.80 |
| | 48–59 | 1.02 | 0.43–2.45 | 0.76 | 0.38–1.55 |
| **Sex of the child** | Male | 1.00 | | 1.00 | |
| | Female | 1.13 | 0.72–1.79 | 1.01 | 0.77–1.33 |
| **Maternal age in years** | <20 | 1.00 | | 1.00 | |
| | 20–29 | 0.76 | 0.43–1.34 | 0.85 | 0.53–1.38 |
| | 30 and above | 0.85 | 0.36–2.04 | 0.71 | 0.35–1.46 |
| **Religion** | Hindu | 1.00 | | 1.00 | |
| | Other | 1.29 | 0.69–2.40 | 1.19 | 0.72–1.95 |
| **Marginalization status** | Triple | 1.00 | | 0.42* | 0.19–0.90 |
| | Double | 1.56 | 0.76–3.21 | 0.42** | 0.24–0.73 |
| | Single | 1.50 | 0.75–2.99 | 0.56* | 0.34–0.92 |
| | No | 1.25 | 0.52–3.02 | 1.00 | |
| **Province** | Koshi | 1.00 | | 1.00 | |
| | Madhesh | 1.22 | 0.45–3.33 | 2.91** | 1.47–5.75 |
| | Bagmati | 0.69 | 0.34–1.39 | 0.81 | 0.44–1.46 |
| | Gandaki | 1.07 | 0.38–3.02 | 0.8 | 0.39–1.64 |
| | Lumbini | 0.57 | 0.27–1.21 | 1.10 | 0.60–2.01 |
| | Karnali | 0.74 | 0.31–1.77 | 0.82 | 0.46–1.47 |
| | Sudurpashchim | 0.56 | 0.27–1.19 | 0.72 | 0.39–1.34 |
| **Residence** | Urban | 1.00 | | 1.00 | |
| | Rural | 0.88 | 0.57–1.36 | 0.86 | 0.62–1.20 |
| **Native language** | Nepali | 1.00 | | 1.00 | |
| | Maithili | 0.63 | 0.23–1.70 | 0.82 | 0.52–1.30 |
| | Bhojpuri | 0.46 | 0.16–1.33 | 0.72 | 0.37–1.41 |
| | Other | 0.78 | 0.43–1.43 | 0.96 | 0.36–2.57 |
| **Birth order** | First | 0.49** | 0.3–0.78 | 1.00 | |
| | Second | 1.00 | | 1.13 | 0.77–1.65 |
| | Third or higher | 0.60 | 0.33–1.08 | 1.22 | 0.72–2.07 |

*Note: Significance at * p < 0.05, ** p < 0.01.*

## Discussion

This study reveals concerning trends regarding common childhood illnesses, such as diarrhea and fever, in Nepal. While the prevalence of these illnesses has increased, there has been a decline in the number of parents seeking healthcare for their children, particularly at public HFs, where services are free of cost across all facility types. In contrast, there is a noticeable shift toward private HFs, with more families seeking care at these institutions, even for services that are available free of charge at public HFs. The study also identified key determinants influencing care-seeking behaviors, including provincial disparities, such as higher rates of diarrhea in Karnali and Bagmati provinces, as well as birth order,

**Table 4. Determinants of children under five years old who had diarrhea, fever and sought care in private HFs, NDHS 2022.**

| Characteristics | Categories | Diarrhea | | Fever | |
|---|---|---|---|---|---|
| | | AOR | 95%CI | AOR | 95%CI |
| **Child age in months** | <6 | 1.00 | | 1.00 | |
| | 6–11 | 2.29 | 0.85–6.15 | 1.15 | 0.48–2.79 |
| | 12–23 | 2.04 | 0.74–5.65 | 1.10 | 0.49–2.45 |
| | 24–35 | 0.92 | 0.30–2.87 | 0.66 | 0.31–1.42 |
| | 36–47 | 1.36 | 0.50–3.65 | 0.93 | 0.43–2.01 |
| | 48–59 | 1.07 | 0.34–3.36 | 0.61 | 0.26–1.44 |
| **Sex of the child** | Male | 1.00 | | 1.00 | |
| | Female | 1.17 | 0.59–2.33 | 0.87 | 0.59–1.29 |
| **Maternal age in years** | <20 | 1.00 | | 1.00 | |
| | 20–29 | 1.13 | 0.43–2.95 | 0.66 | 0.38–1.15 |
| | 30 and above | 0.62 | 0.16–2.41 | 1.19 | 0.54–2.59 |
| **Religion** | Hindu | 1.00 | | 1.00 | |
| | Other | 0.69 | 0.26–1.84 | 1.64 | 0.88–3.07 |
| **Marginalization status** | Triple | 1.00 | | 1.00 | |
| | Double | 0.73 | 0.21–2.61 | 1.61 | 0.73–3.57 |
| | Single | 1.21 | 0.35–4.18 | 4.58*** | 2.03–10.3 |
| | No | 2.45 | 0.54–11.18 | 10.98*** | 4.22–28.57 |
| **Province** | Koshi | 0.95 | 0.30–3.01 | 1.07 | 0.49–2.35 |
| | Madhesh | 1.78 | 0.46–6.93 | 3.64** | 1.52–8.72 |
| | Bagmati | 2.15 | 0.55–8.48 | 0.98 | 0.43–2.23 |
| | Gandaki | 0.87 | 0.15–5.06 | 1.80 | 0.68–4.75 |
| | Lumbini | 3.48* | 1.05–11.55 | 4.69*** | 2.06–10.67 |
| | Karnali | 0.46 | 0.19–1.14 | 0.56 | 0.28–1.12 |
| | Sudurpashchim | 1.00 | | 1.00 | |
| **Residence** | Urban | 1.00 | | 2.28*** | 1.48–3.5 |
| | Rural | 0.66 | 0.32–1.34 | 1.00 | |
| **Native language** | Nepali | 1.00 | | 1.00 | |
| | Maithili | 4.38* | 1.15–16.76 | 5.65*** | 2.39–13.34 |
| | Bhojpuri | 1.74 | 0.28–10.89 | 1.63 | 0.60–4.43 |
| | Other | 3.06* | 1.24–7.56 | 1.82* | 1.00–3.29 |
| **Birth order** | First | 1.00 | | 1.00 | |
| | Second | 0.93 | 0.45–1.92 | 1.63 | 1.00–2.68 |
| | Third or higher | 1.61 | 0.54–4.86 | 0.93 | 0.51–1.72 |

*Note: Significance at * p < 0.05, ** p < 0.01, *** p < 0.001.*

with first-born children being less likely to seek treatment. Furthermore, the language spoken at home emerged as a significant factor affecting access to care for childhood illnesses. These trends point to considerable inequities in healthcare access, with disadvantaged groups facing greater barriers to care, highlighting the urgent need for targeted interventions to address these disparities.

Overall, the prevalence of fever has risen from 19% in 2011 to 21% in 2016 and 23% in 2022, and the prevalence of diarrhea has risen from 8% in 2016 to 10% in 2022. These illnesses are just the tip of the iceberg, with broader spill-over effects extending to sectors beyond health, including water and sanitation, waste management, housing and living

standards, and health literacy. The increasing prevalence of these preventable diseases and signs of fever highlight underlying issues such as poor water and sanitation, as well as poor housing and living conditions [38,39]. More than 80% of the underlying factors of communicable diseases are caused by unsafe water, unhygienic food, and environmental factors [40].The rising trends of these preventable and common childhood illnesses, such as diarrhea, are the manifestation of several waterborne and foodborne conditions [41]. These issues are a major public health concern and these social determinants of health remain a neglected paradigm in Nepal [41]. In contrast, fever manifests in several childhood infections, especially acute respiratory infections. Potential reasons for rising prevalence of fever could be the increased cases due to perceived severity and awareness during the COVID-19 pandemic. Studies from other low- and middle-income countries especially from Africa and Asia also revealed that the prevalence of fever is higher than in Nepal [42,43].

Childhood illnesses such as diarrhea and fever manifest through poor living conditions, personal hygiene, socioeconomic status, waste management, water and sanitation, indoor and outdoor pollution, and nutrition, particularly among children who are highly susceptible to the disease conditions. Current child health programs are more focused on treating diseases rather than addressing the root causes of childhood illnesses [44]. Families and communities consider health care as the provision of medicine, doctors, and HFs for people who are sick rather than focusing on health promotion and prevention through multisectoral policies and actions. The increased prevalence of childhood illnesses is a burden to the health system [44]. The Global Burden of Diseases Report estimated that around 20–30% of total children under five years old in Nepal suffer from childhood diarrhea annually [45], while an estimated 10% of deaths are caused by diarrhea [46]. Additionally, fever is the first symptom of infection or malnutrition, including respiratory or intestinal infections [43]. In many LMICs, infants and younger children suffer more from ARIs, which are associated with poor housing, lack of ventilation in housing, indoor air pollution, and passive smoking [47].

The younger children aged six months to four years were more likely to suffer from fever, while children aged three to five years had a higher prevalence of diarrhea and children who were second in the birth order and from Karnali province had a higher prevalence of childhood illnesses, particularly diarrhea Children from disadvantaged families (Karnali, second birth order, Hill regions, and Maithili speakers) had a higher prevalence of these conditions. The higher prevalence of diarrhea among older children may be due to increased mobility once they begin walking, which exposes them to more sources of infection as they interact with their surroundings [48]. Water, sanitation, and personal hygiene require better attention at this stage [49]. In families with higher birth order or in certain provinces, lower awareness and health literacy could be the causes of diarrhea among those groups.

Overall trends of treatment-seeking showed a declining trend for both diarrhea and fever in the last two surveys from 2016 and 2022. Among those seeking care, there is a decline in care-seeking from public HFs. The NHFS 2021 reported that access to HF-based childhood services has increased in Nepal [50]. However, this study reported a decrease in overall care seeking practices for childhood illnesses, indicating that many children are not receiving care, even though they are ill. The proportion of no treatment was higher among children from disadvantaged families than those of children from privileged groups. Poor treatment-seeking practices, where affected children are not receiving necessary care or medical advice, indicate that many children are suffering from preventable illnesses. This lack of treatment may be a key factor contributing to higher morbidity and mortality rates, particularly among children from disadvantaged families and communities. Children from disadvantaged groups with low care-seeking for childhood illnesses mean either the group had not reached HFs for treatment or lacked trust in public HFs and, therefore, did not seek care [25].

There was overall decline in care-seeking from public HFs, especially among disadvantaged groups. During the COVID-19 pandemic, the delivery and care-seeking for routine services were hampered, such as when children missed their vaccinations [51]. However, the current care-seeking practice indicates that the CB-IMNCI program needs to perform better in delivering health service interventions to children with illnesses and indicates areas for improvement. Communities have poor trust in public HFs and care providers, limited availability of essential drugs, lack of uninterrupted health services in HFs, or suboptimal quality of care [52]. The GON has invested significant amount in the CB-IMNCI program,

but low treatment-seeking in public HFs undermines the program's ability to reach more children and play a larger role in treatment of childhood illnesses [53].

Among those who sought care, nearly two-thirds (73%) of children with diarrhea and four in five (80%) children with fever sought treatment at private HFs. Data from the NHFS 2021 shows universal child health service availability but readiness and compliance to standards of care are poor [54]. including lack of having trained staff in HFs, and shortage of essential medicines and supplies for childhood illnesses [54]. Many parents perceive that public HFs provide poor-quality care, substandard medications, or delayed treatment for childhood illnesses [55]. As a result, mothers are reluctant to risk their children's health and prefer to seek care at private HFs. This trend is further compounded by the presence of privately owned pharmacies close to public HFs, where over-the-counter medications are readily available. The choice to seek treatment in private settings is influenced by the desire for immediate care, both from the perspective of healthcare providers and mothers. Earlier evidence from western Nepal reported that, of those who sought care, two in five visited private pharmacies directly instead of public HFs, particularly when children had a fever [25].

Despite the availability of free treatment for childhood illnesses through the CB-IMNCI program in public HFs, a higher proportion of children continue to seek care in the private sector, and this trend is on the rise. The increasing reliance on private HFs has several important policy implications. First, in private HFs, health services, including BHS, are pay-for-service, which could result in additional financial burdens for families and communities. Second, although providers in selected private HFs in selected districts are trained on CB-IMNCI, this has not been nationwide and the monitoring mechanism is weak, and healthcare providers in these settings may have limited awareness of the CB-IMNCI treatment protocols [14]. As a result, the treatment of childhood illnesses in private HFs may not align with national standards. Third, private HFs in Nepal are often poorly regulated, lacking accreditation and effective monitoring mechanisms. Without adequate oversight, healthcare providers in private HFs may engage in practices such as overtreatment and over-prescription, exceeding the guidelines set by national treatment standards [56]. In many cases, healthcare providers in private HFs prescribe antibiotics that go beyond the CB-IMNCI treatment protocol, order unnecessary laboratory tests, and respond to caretakers who request and demand such advanced services. Such overprescription practices not only increase financial burdens for disadvantaged and socially excluded families but also contribute to the growing problem of antibiotic resistance, particularly in the treatment of common childhood illnesses. Furthermore, certain underprivileged groups, including those from the Lumbini and Madhesh provinces of the Terai region, native Maithili speakers, residents of urban areas, and younger children, exhibited higher rates of care-seeking in private HFs. This trend may be attributed to a sense of urgency, a lack of trust in government healthcare systems, and the perception of inadequate or delayed treatment and care in public HFs.

There is a trend of limited care-seeking practices for fever and diarrhea in Nepal, but the reasons behind this remain poorly understood. However, potential reasons for not seeking care could be multiple. For instance, although care services for childhood illnesses are free, the perceived quality of public facilities is low. Mothers and caregivers of children often bypass local facilities and choose private medical centers or pharmacies for treatment of such childhood illnesses [25]. While the perceived quality of services in private facilities is higher, the cost of care in these facilities is also high. Common childhood illnesses are episodic, requiring frequent care throughout the year [24]. Households often face economic barriers when accessing healthcare, and as a result, they sometimes manage symptoms with home care interventions [25]. Another reason could be that parents have a low perception of the risks associated with childhood illnesses and believe the symptoms will subside on their own.

Despite progress in expanding healthcare services, childhood illnesses like diarrhea and fever remain inadequately treated in Nepal due to multiple challenges. Challenges with health system readiness such as the availability of trained staff, ORS, or child-appropriate medicines remain limited, particularly in both public and private facilities. National health system capacity, including physical infrastructure and human resources, has improved only marginally over the last two decades and has not kept pace with the rising burden of childhood illnesses [53]. Moreover, persistent supply chain

challenges such as stock-outs, difficult terrain, and reliance on imports continue to hinder consistent availability of treatment at service points. Furthermore, climate change has been linked to an increase in the frequency and intensity of diarrheal diseases, while increased care costs at the households level, particularly during the monsoon and warmer months, placing additional strain on already overstretched health services and widening the gap between need and access [57,58]. Climate change has increased the burden of disease, especially among vulnerable children, by worsening waterborne and vector-borne infections. While the Universal Health Coverage service index rose from 20% to 54% between 2000 and 2021—showing gains in maternal, child, and infectious disease services, the overall health system capacity, including workforce and hospital beds, improved only marginally (from 25% to 37%) [53,58]. Access to care remains uneven, particularly in rural areas, and many families face high indirect costs such as lost wages, making timely treatment unaffordable. These structural gaps in service delivery may reinforce caregivers' perceptions that formal care is either inaccessible or ineffective. Nonetheless, these are important findings from this study that warrant further exploration and investigation to understand why mothers and caregivers choose not to seek care for common childhood illnesses.

## Limitations of study

This study has several limitations. First, it included mothers who had children under five years preceding the survey and whose children had experienced childhood illnesses in the two weeks before the survey. We constructed a composite disadvantage measure from education, wealth status, and ethnicity. Due to small sample sizes, particularly among children with one or two forms of disadvantage, some variables were combined for analysis, such as merging categories of marginalization status. Care-seeking behavior for acute respiratory infections (ARI) was excluded because of the small number of ARI cases (n = 73). Additionally, while the cost of care is an important determinant, especially since families often pay for care in private HFs—data limitations prevented analyzing who bore the costs of care for diarrhea and fever in public HFs. Further limitations include cross-sectional study design, reliance on self-reported data which may introduce recall bias, and the absence of quality-of-care indicators in the analysis. Future surveys like the NDHS could address these limitations by including relevant questions and indicators to improve data accuracy and comprehensiveness.

## Implications for policy programs and research

Based on the findings, the following policy and program recommendations are drawn. First, Awareness and health information should prioritize reducing the prevalence of childhood diseases, especially in Karnali province, among those with a higher birth order than one, younger groups, and disadvantaged groups. Designing and implementing programs is essential to target marginalized groups and coordinate with local governments, including the provision of health education, information, and communication programs in local languages. Second, the CB-IMNCI program needs to be assessed, particularly focusing on investment and outputs (in terms of service delivery and care-seeking). Third, Care-seeking practices need to improve, particularly in public HFs, including the provision of medicines, trained health workers, and uninterrupted availability of health services. Recruitment of health workers who speak local languages in the Terai region is also essential. Local health systems and care providers should work closely with the community, and FCHVs could identify children with fevers and refer them to HFs. Fourth, families and communities should be able to choose where to seek care for their children. The government should ensure that care and treatment facilities in public HFs are adequate and develop the public's trust in the system. Policies on regulating and monitoring private providers through prescription-based care practices are essential. Trust in the public health system and the quality of healthcare delivery need to be improved by strengthening the health system and increasing the supply of essential medicines and trained health workers. The CB-IMNCI Program has not been implemented in most private HFs[44]. Fifth, addressing these factors requires multisectoral actions, especially from beyond the health sector, including water and sanitation, education for improved health literacy on the value of nutritional diets, hygiene, reducing indoor pollution, and vaccinations (for instance, rotavirus) [59].

Prevention is better than a cure, so multisectoral policies and actions need to focus on preventing and lowering the prevalence of these diseases [59]. Government policies should focus on improving the underlying factors of these diseases rather than on investment in treatment. Finally, future studies could investigate why mothers visit private HFs to treat childhood illnesses, including how much the cost was incurred, and a study on the use of antibiotics in treating childhood illnesses would be informative. In the context of increased no-treatment and poor care-seeking in public HFs, future studies on the underlying factors behind these trends are warranted.

## Conclusions

The prevalence of childhood fever and diarrhea has been increasing, particularly in recent years. These two common childhood illnesses serve as indicators of the overall effectiveness of the health system's child health programs. While the overall care-seeking practices have declined, there has been a notable shift toward increased reliance on private HFs. The trends and determinants identified in the study also highlight significant equity gaps, with disadvantaged groups facing greater challenges in accessing care. Policy formulation, program implementation, and service delivery must prioritize children from structurally disadvantaged backgrounds, addressing the underlying social determinants of health by improving living conditions, housing conditions, health-related behaviors, and awareness through cross-sectoral policies and interventions beyond the health sector. Reducing common childhood illnesses including diarrhea and fever and improving care seeking practice could significantly reduce preventable childhood illnesses, particularly among children from mothers of socioeconomically and geographically disadvantaged backgrounds.

## Supporting information

**S1 Table. Prevalence of diarrhea and fever among children under five two weeks prior to the survey, NDHS 2022.**
(DOCX)

**S2 Table. Distribution of children under five years old who sought care for the treatment of diarrhea, and fever, NDHS 2022.**
(DOCX)

**S3 Table. Distribution of children under five years old who had diarrhea two weeks prior to the survey and sought the treatment of diarrhea, and fever private HFs, NDHS 2022.**
(DOCX)

**S4 Table. Bivariable logistic regression analysis of children under 5 who had diarrhea and fever 2 weeks prior to the survey, NDHS 2022.**
(DOCX)

**S5 Table. Bivariable logistic regression analysis of children under 5 who had diarrhea and fever and sought care in the 2 weeks prior to the survey, NDHS 2022.**
(DOCX)

**S6 Table. Bivariable logistic regression analysis of children under 5 who had diarrhea, and fever and sought care in private HFs in the 2 weeks prior to the survey, NDHS 2022.**
(DOCX)

## Author contributions

**Conceptualization:** Resham B Khatri, Sabita Tuladhar.

**Data curation:** Resham B Khatri.

**Formal analysis:** Resham B Khatri.

**Investigation:** Resham B Khatri, Rolina Dhital, Sabita Tuladhar, Ravi Kanta Mishra, Yibeltal Assefa.

**Methodology:** Resham B Khatri, Sabita Tuladhar.

**Project administration:** Resham B Khatri.

**Resources:** Sabita Tuladhar.

**Software:** Resham B Khatri.

**Supervision:** Yibeltal Assefa.

**Validation:** Resham B Khatri, Rolina Dhital, Sabita Tuladhar, Ravi Kanta Mishra, Yibeltal Assefa.

**Visualization:** Resham B Khatri, Sabita Tuladhar.

**Writing – original draft:** Resham B Khatri, Rolina Dhital.

**Writing – review & editing:** Resham B Khatri, Rolina Dhital, Sabita Tuladhar, Ravi Kanta Mishra, Yibeltal Assefa.

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
