## [Decision Letter · Decision Letter 0]

12 Aug 2025

PGPH-D-25-01861

Trends and determinants of prevalence and care-seeking of diarrhea and fever among children under five in Nepal: an analysis of the most recent Nepal Demographic and Health Surveys

Dear Dr. Khatri,

Thank you for submitting your manuscript to PLOS Global Public Health. After careful consideration, we feel that it has merit but does not fully meet PLOS Global Public Health’s publication criteria as it currently stands. Therefore, we invite you to submit a revised version of the manuscript that addresses the points raised during the review process.

Please note that we have only been able to secure a single reviewer to assess your manuscript. We are issuing a decision on your manuscript at this point to prevent further delays in the evaluation of your manuscript. Please be aware that the editor who handles your revised manuscript might find it necessary to invite additional reviewers to assess this work once the revised manuscript is submitted. However, we will aim to proceed on the basis of this single review if possible. 

The reviewer has highlighted some concerns with the methodological reporting of this manuscript. Please ensure the manuscript includes all details required to replicate the study. We also note that some references have been suggested, as always, we invite you to review the relevance of these to your work but there is no requirement to include them. 

We look forward to receiving your revised manuscript.

Kind regards,

Joanna Tindall, PhD

Staff Editor

Journal Requirements:

Please send a completed 'Competing Interests' statement, including any COIs declared by your co-authors. If you have no competing interests to declare, please state "The authors have declared that no competing interests exist".

Please amend your detailed Financial Disclosure statement. This is published with the article. It must therefore be completed in full sentences and contain the exact wording you wish to be published.If you did not receive any funding for this study, please simply state: “The authors received no specific funding for this work.”

Please provide separate figure files in .tif or .eps format.For more information about figure files please see our guidelines:https://journals.plos.org/globalpublichealth/s/figureshttps://journals.plos.org/globalpublichealth/s/figures#loc-file-requirements

We have noticed that you have uploaded Supporting Information files, but you have not included a list of legends. Please add a full list of legends for your Supporting Information files after the references list.

We note that your Data Availability Statement is currently as follows: “Data is provided within the manuscript or supplementary information files.”Please confirm at this time whether or not your submission contains all raw data required to replicate the results of your study. Authors must share the “minimal data set” for their submission. PLOS defines the minimal data set to consist of the data required to replicate all study findings reported in the article, as well as related metadata and methods (https://journals.plos.org/plosone/s/data-availability#loc-minimal-data-set-definition).For example, authors should submit the following data:- The values behind the means, standard deviations and other measures reported;- The values used to build graphs;- The points extracted from images for analysis.Authors do not need to submit their entire data set if only a portion of the data was used in the reported study.If your submission does not contain these data, please either upload them as Supporting Information files or deposit them to a stable, public repository and provide us with the relevant URLs, DOIs, or accession numbers. For a list of recommended repositories, please see https://journals.plos.org/plosone/s/recommended-repositories.If there are ethical or legal restrictions on sharing a de-identified data set, please explain them in detail (e.g., data contain potentially sensitive information, data are owned by a third-party organization, etc.) and who has imposed them (e.g., an ethics committee). Please also provide contact information for a data access committee, ethics committee, or other institutional body to which data requests may be sent. If data are owned by a third party, please indicate how others may request data access.

Additional Editor Comments (if provided):

Reviewers' comments:

Reviewer's Responses to Questions

**Comments to the Author**

1. Does this manuscript meet PLOS Global Public Health’s publication criteria?

Reviewer #1: Yes

2. Has the statistical analysis been performed appropriately and rigorously?

Reviewer #1: Yes

3. Have the authors made all data underlying the findings in their manuscript fully available (please refer to the Data Availability Statement at the start of the manuscript PDF file)?

Reviewer #1: Yes

4. Is the manuscript presented in an intelligible fashion and written in standard English?

Reviewer #1: Yes

Reviewer #1: Minor comments:

Abstract

1. Line 24: Incomplete sentence

2. I suggest that the author add a summary of the statistical analysis applied in this study.

Results

The inclusion of six figures and four tables makes the manuscript feel quite extensive. The authors may consider streamlining or consolidating some of the figures and tables to enhance clarity and conciseness.

Figures

3. Statistically significant signs should be in all figures' footnotes.

Typo

4. Line 495: “Fifthly, addressing Addressing”

Discussion

5. What is the reason for the increasing percentage of "Not treated" diarrhea in Nepal? Please add an explanation in the discussion section.

Major Comments:

Introduction

6. The research or knowledge gaps and aims should be more clearly explained by reviewing the existing literature on this topic, particularly studies conducted in Nepal. For example, previous studies have investigated Trends and prevalence of diarrhea in Nepal, such as:

a. Paudel, D., et al. (2020): Determinants of childhood diarrhoea in Nepal: A multilevel analysis of Nepal Demographic and Health Survey 2016. PLoS ONE, 15(11), e0242570.

https://doi.org/10.1371/journal.pone.0242570

b. Ghimire, U., et al. (2018): Trends and inequalities in diarrhoea among children under five years in Nepal: 2001–2016. Tropical Medicine & International Health, 23(10), 1050–1060. https://doi.org/10.1111/tmi.13127

Please elaborate on what has already been done in Nepal in this field and clearly identify the remaining research gaps that this study seeks to address.

Method

7. This study has not clearly explained how the trend analysis was performed, i.e., which trend test was utilized?

8. I would like to suggest that the author apply a mixed-effect modeling approach in this study. Demographic Health Survey (DHS) data is hierarchical. Individuals are nested within households, which are nested within clusters, which are nested within regions. Mixed effects models allow for correlated observations within clusters or groups by including random effects.

**Do you want your identity to be public for this peer review?** For information about this choice, including consent withdrawal, please see our Privacy Policy

Reviewer #1: No

---

## [Decision Letter · Decision Letter 1]

17 Oct 2025

PGPH-D-25-01861R1

Rising childhood illnesses (diarrhea and fever) and shifting care-seeking practices in Nepal: insights from three most recent demographic and health surveys (2011, 2016 and 2022)

Dear Dr. Khatri,

Thank you for submitting your manuscript to PLOS Global Public Health. After careful consideration, we feel that it has merit but does not fully meet PLOS Global Public Health’s publication criteria as it currently stands. Therefore, we invite you to submit a revised version of the manuscript that addresses the points raised during the review process.

We look forward to receiving your revised manuscript.

Kind regards,

Dickson Abanimi Amugsi, PhD

Academic Editor

Journal Requirements:

Additional Editor Comments (if provided):

Reviewer #1:

Reviewers' comments:

Reviewer's Responses to Questions

**Comments to the Author**

Reviewer #1: All comments have been addressed

publication criteria?

Reviewer #1: Yes

3. Has the statistical analysis been performed appropriately and rigorously?

Reviewer #1: Yes

4. Have the authors made all data underlying the findings in their manuscript fully available (please refer to the Data Availability Statement at the start of the manuscript PDF file)?

Reviewer #1: Yes

5. Is the manuscript presented in an intelligible fashion and written in standard English?

Reviewer #1: Yes

Reviewer #1: Thank you so much, authors, for addressing all of my comments.

**Do you want your identity to be public for this peer review?** For information about this choice, including consent withdrawal, please see our Privacy Policy

Reviewer #1: No

---

## [Decision Letter · Decision Letter 2]

2 Dec 2025

Rising childhood illnesses (diarrhea and fever) and shifting care-seeking practices in Nepal: insights from three most recent demographic and health surveys (2011, 2016 and 2022)

PGPH-D-25-01861R2

Dear Dr. Khatri,

We are pleased to inform you that your manuscript 'Rising childhood illnesses (diarrhea and fever) and shifting care-seeking practices in Nepal: insights from three most recent demographic and health surveys (2011, 2016 and 2022)' has been provisionally accepted for publication in PLOS Global Public Health.

Best regards,

Dr Tanmay Bagade, Ph.D., MS (O&G), MPH, MHM

Academic Editor

Reviewer Comments (if any, and for reference):

Reviewer's Responses to Questions

**Comments to the Author**

Reviewer #1: All comments have been addressed

publication criteria?

Reviewer #1: Yes

3. Has the statistical analysis been performed appropriately and rigorously?

Reviewer #1: Yes

4. Have the authors made all data underlying the findings in their manuscript fully available (please refer to the Data Availability Statement at the start of the manuscript PDF file)?

Reviewer #1: Yes

5. Is the manuscript presented in an intelligible fashion and written in standard English?

Reviewer #1: Yes

Reviewer #1: Thank you so much for previously addressing all of my comments. I don't have further comments.

**Do you want your identity to be public for this peer review?** For information about this choice, including consent withdrawal, please see our Privacy Policy

Reviewer #1: No
